# GUARANTEED ADAPTIVE-K IN RECOMMENDATIONS

## ABSTRACT

Recommender systems (RS) are crucial in offering personalized suggestions tailored to user preferences. While conventionally, Top-$K$ recommendation approach is widely adopted, its reliance on fixed recommendation sizes overlooks the diverse needs of users, leading to some relevant items not being recommended or vice versa. While recent work has made progress, they determine $K$ by searching over all possible recommendation sizes for each user during inference. In real-world scenarios, with large datasets and numerous users with diverse and extensive preferences, this process becomes computationally impractical. Moreover, there is no theoretical guarantee of improved performance with the personalized K. In this paper, we propose a novel framework, **K-Adapt**, which determines dynamic K-prediction set size for each user efficiently and effectively. Specifically, it reformulates adaptive Top-K recommendation as a utility-based risk control problem, where a calibrated threshold based on user utility metrics determines the prediction sets. A lightweight greedy optimization algorithm efficiently learns this threshold to generate dynamic recommendations. Theoretical analysis is provided by establishing upper bounds on expected risk as well as near-optimality and stability of learned threshold. Extensive experiments on multiple datasets demonstrate that K-Adapt framework outperforms baseline methods in both performance and time efficiency, offering a guaranteed solution to fixed Top-K challenges.

## 1 INTRODUCTION

With the growing relevance of the web as a medium for electronic and commercial transactions, Recommender Systems (RS) (Isinkaye et al., 2015; Lu et al., 2015; Aggarwal, 2016; Zhao et al., 2023) have become ubiquitous for mitigating information overload, enabling platforms to deliver relevant suggestions across various domains such as e-commerce (Gulzar et al., 2023), entertainment (Perano et al., 2021), and job matching (Islam et al., 2021). They rank items based on users' preferences and their historical behaviors (Khatwani & Chandak, 2016; Cui et al., 2020), thereby presenting the Top-K items (Cremonesi et al., 2010; Li et al., 2020; Wei et al., 2024), with the ranking scores, sorted in descending order. While this heuristic Top-$K$ recommendation approach is widely adopted in the literature for its simplicity, a fundamental limitation is often overlooked: its reliance on a fixed $K$. This approach assumes that the same recommendation size will suffice for all users, ignoring their diverse needs and leading to some relevant items not being recommended or vice versa. As a result, poor recommendation performance across users can lead to dissatisfaction and disengagement from the platform (Chen et al., 2022). For example, Figure 1 illustrates how the NeuMF model's oracle performance on the Last.fm dataset obtained by dynamically selecting each user's best per-metric set size (capped at 25) differs substantially from the performance under a single, fixed $k$ ( where fixed $k$ is derived by averaging the user-specific (oracle) set sizes across the dataset for each metric) highlighting the inherent weakness of fixed-$K$ recommendations.

While the concept of dynamically tailoring $K$ to individual users is promising across multiple recommendation settings like optimizing screen space (Xi et al., 2023), balancing user engagement and budget constraints (Chen et al., 2022), or reducing user overload, existing research in this direction remains very limited. Recently, KWEON et al. (2024) modeled user-item interactions using Bernoulli distributions during inference and approximated utility over ranked lists with Poisson-Binomial distribution to determine optimal $K$ for each user. However, evaluating utility during inference is computationally impractical for real-world systems handling extra high dimension $K$, millions of users, and numerous preferences. This chal-

---

The code and implementation details are available at https://anonymous.4open.science/r/Top-Adaptive-K-551B

lenge also shares parallels with document list truncation methods (Wu et al., 2021; Wang et al., 2022). However, these methods are prone to overfitting and poor generalization in the sparse and noisy contexts of recommendation datasets. Moreover, none of these methods provides statistical guarantees for model performance, which is essential for trustworthy recommendations.

Motivated by the above-mentioned challenges and taking inspiration from Conformal Prediction (CP) (Schafer et al., 1999; Vovk et al., 2005; Fontana et al., 2023), we aim to propose a statistically sound user-tailored framework that incorporates uncertainty quantification into the RS ecosystem. Specifically, our framework aims to achieve two key objectives: (1) dynamically and efficiently determining prediction set size for each user, and (2) statistically ensuring that these dynamically generated prediction set sizes meet the desired performance guarantees across the dataset. However, CP in its classical form is not suitable for this setting as: (a) it does not align with the goals of recommender systems, where

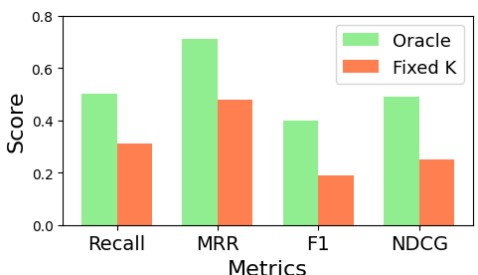

Figure 1: Comparison of Oracle vs. Fixed $K$ on Last.fm dataset using NeuMF model.

performance is measured by ranking-based utilities such as Recall or NDCG rather than simple label inclusion; and (b) its guarantees are marginal and label-oriented, offering no direct control over user-level risk with respect to these utilities. As a result, naive application of CP will result in overly conservative recommendation sets that sacrifice relevance and user satisfaction.

To address these challenges, we propose **K-Adapt**, a dataset- and model-agnostic statistical framework that frames adaptive top-$K$ recommendation as a risk-controlled prediction problem. Instead of guaranteeing label coverage as in classical CP, K-Adapt defines user-level loss functions based on ranking utilities and constrains their expected risk below a user-defined threshold $\alpha$ with confidence $1 - \eta$. Specifically, K-Adapt leverages the output scores of a base recommender and applies a calibrated threshold parameter $\lambda$ to determine items recommended to each user. This calibration balances set size and utility, producing recommendation lists that are compact and reliable. Importantly, by operating as a post-hoc calibration layer for the truncation stage, K-Adapt avoids the conservativeness of classical CP while optimizing the critical trade-off between candidate coverage and display constraints under statistical validity. Our contributions are as follows:

- We first propose a novel framework, K-Adapt, which reformulates adaptive recommendation as a utility-based risk control problem extending the CP paradigm, and defines loss functions tailored to key RS performance metrics.

- Secondly, we develop a light-weight greedy-based optimization algorithm to efficiently calibrate the threshold $\lambda$ and ensure that the utility-based risk remains below a user-defined target.

- Next, we provide rigorous theoretical analysis showing that the calibrated threshold $\hat{\lambda}$ not only controls the expected utility-based risk near the user-specified level $\alpha$ with high probability (Theorem 1), but also lies near the population-optimal threshold (Theorem 2) and remains stable under sampling perturbations of the calibration set (Theorem 3).

- Finally, we conduct extensive experiments across multiple datasets and metrics (Section 5) to demonstrate the effectiveness of K-Adapt in both performance as well as time efficiency.

## 2 RELATED WORKS

### 2.1 PERSONALIZED RECOMMENDATION SIZE

In RS, the most common practice involves recommending top-K fixed item for each user (Yang et al., 2012; Kweon et al., 2021; Kang et al., 2022; Li et al., 2024). The idea of dynamically tailoring the recommendation set size to individual users' preferences is a novel research direction that has received limited attention. It draws parallels with the document list truncation problem, which determines the optimal cutoff position for retrieved documents (Arampatzis et al., 2009; Wu et al., 2021). Some recent methods such as AttnCut and MtCut (Bahri et al., 2020; Wang et al., 2022) utilize deep models to frame this truncation task as a classification problem to predict the optimal cutoff position using a $K$-dimensional probability target vector. Recently, KWEON et al. (2024) proposed PerK, which leverages calibrated interaction probabilities to estimate expected user utility and select optimal personalized recommendation sizes.

## 2.2 Conformal Prediction

Conformal prediction (Papadopoulos et al., 2002; Shafer & Vovk, 2008; Angelopoulos & Bates, 2022), provides finite-sample, distribution-free guarantees by constructing prediction sets that ensure the coverage guarantee $P(Y \notin C(X)) \leq \alpha$. This foundational method is model-agnostic and has been widely adopted in applications requiring robust uncertainty quantification.

Recent advancements in conformal methods have extended these guarantees to address more complex challenges. Works like Tibshirani et al. (2019) explore conformal prediction under distributional shifts, while Bates et al. (2021) introduce high probability risk bounds to control errors beyond miscoverage. Building on these advancements, conformal risk control (Angelopoulos et al., 2024) generalizes conformal prediction to guarantee the expected value of monotone functions, expressed as $E[\ell(C_\lambda(X), Y)] \leq \alpha$. Here, $\lambda$ is tunable parameter to balance prediction set size $C_\lambda(X)$ and controlled loss and $\alpha$ is desired error rate. This framework expands conformal prediction's utility, enabling applications like false negative rate control in settings like multilabel classification.

## 3 The Proposed Framework

We begin by introducing notations used in this paper. Consider $m$ items, denoted as $\boldsymbol{i} = [i]_{j=1}^m$, where each item $i_j$ is an element of the item space $\mathcal{I}$. Similarly, we have $n$ users, represented by $\boldsymbol{u} = [u]_{k=1}^n$, where each user $u_k$ belongs to the user space $\mathcal{U}$. For brevity, we use $u$ and $i$ to denote a user and an item, respectively.

We focus on the recommendation with implicit feedback (Hu et al., 2008; He et al., 2016; Zhu et al., 2024), a widely adopted scenario in RS. For a pair $(u, i)$, an interaction label $Y_{u,i}$ is assigned a value of 1 if the interaction is observed, and 0 otherwise. Note that when $Y_{u,i} = 0$, it indicates that the item $i$ may either be irrelevant to the user $u$ or a hidden-relevant item. These interaction labels are used to define $I_{\text{true}}(u)$, the set of all relevant items for user $u$ i.e. $I_{\text{true}}(u) = \{i \mid Y_{u,i} = 1\}$.

A dataset $\mathcal{D} = \{(u, i) \mid Y_{u,i} = 1\}$ consists of observed positive pairs and is partitioned into three mutually exclusive subsets: training ($\mathcal{D}_{\text{train}}$), calibration ($\mathcal{D}_{\text{calib}}$), and testing ($\mathcal{D}_{\text{test}}$). For a user $u$, the unobserved itemset $I_u^- = \{i \mid (u, i) \notin \mathcal{D}_{\text{train}}\}$ represents all items not observed in the training set. This unobserved itemset is further partitioned into two disjoint subsets: $\mathcal{I}_u^{\text{calib}}$ and $\mathcal{I}_u^{\text{test}}$, corresponding to the calibration phase and testing phase, respectively.

**Prediction Sets** After the recommender model $f_\theta : \mathcal{U} \times \mathcal{I} \rightarrow [0, 1]$ is trained on $\mathcal{D}_{\text{train}}$, we produce a ranked list $\pi(u)$ for the unobserved items in $I_u^-$ by sorting their relevance scores $f_\theta(u, i)$ in descending order:

$$\pi(u) = \text{sort}_{i \in I_u^-} f_\theta(u, i), \tag{1}$$

where $\pi(u)$ represents the ranked order of unobserved items based on the estimated scores. Here, $I_u^-$ is either $\mathcal{I}_u^{\text{calib}}$ or $\mathcal{I}_u^{\text{test}}$, depending on the calibration or testing phase respectively.

**Top-$K$ Predictions** Traditionally, recommender systems generate *Top-$K$ predictions* for a given user $u$ by selecting the $K$-most relevant items from the ranked list $\pi(u)$:

$$\pi_K(u) = \pi(u)[: K], \tag{2}$$

where$[: K]$ denotes selecting the first $K$ elements of the ranked list $\pi(u)$.

While this fixed-$K$ approach is commonly adopted for simplicity, it fails to adapt to user-specific preferences and varying recommendation quality across users. This limitation motivates the exploration of dynamic prediction set sizes to better align recommendations with user needs. To address this, we develop our K-Adapt framework which creates personalized dynamic prediction set sizes for each user to ensure guaranteed performance across different recommendation metrics.

We begin by defining our set predictor dominated by the parameter $\lambda$ to output calibrated prediction set $\pi_\lambda(u)$. The calibrated prediction list $\pi_\lambda(u)$ is given by:

$$\pi_\lambda(u) = [i \in \pi(u) \mid f_\theta(u, i) \geq \lambda], \tag{3}$$

where the items are retained in the same order as in $\pi(u)$.

By construction, the predictor satisfies the following property:

$$\lambda_1 < \lambda_2 \implies \pi_{\lambda_2}(u) \subseteq \pi_{\lambda_1}(u). \tag{4}$$

Next, to quantify the alignment between the calibrated prediction set $\pi_\lambda(u)$ and the ground-truth relevant items $I_{\text{true}}(u)$. We introduce a general utility functional:

$$U_M : 2^\mathcal{I} \times 2^\mathcal{I} \rightarrow [0, 1],$$

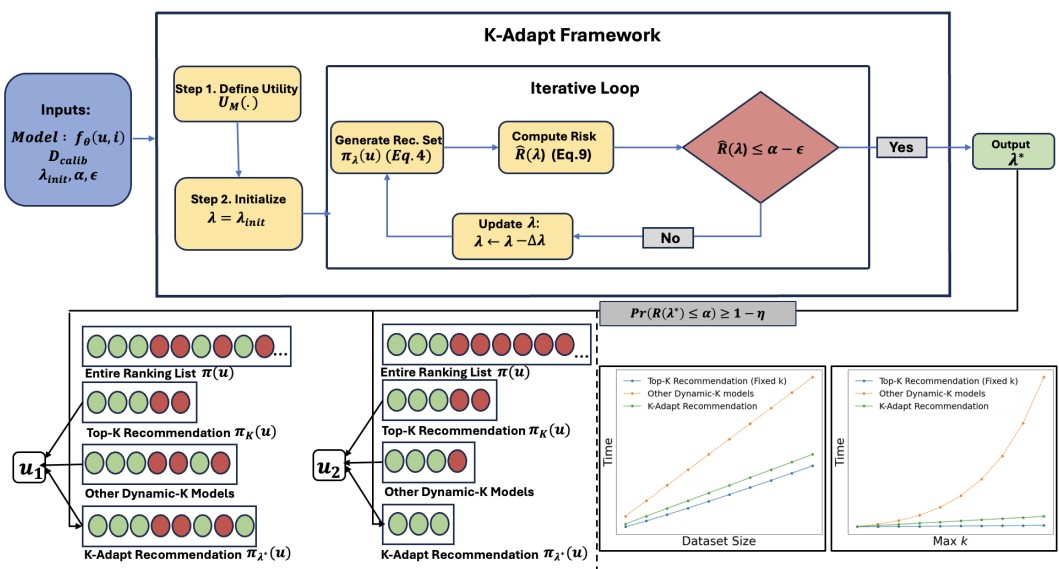

Figure 2: **The K-Adapt Framework.** The *top* portion outlines how $\lambda$ is calibrated to choose recommendation list lengths under a specified risk constraint. In the **lower left**, we compare final recommendations from top-$K$, other dynamic-$K$, and K-Adapt (green = relevant, red = irrelevant), showing how K-Adapt flexibly selects up to a maximum allowable length (max $k$) while maintaining high relevance. The **lower right** plots runtime versus dataset size and max $k$, illustrating that K-Adapt's computational overhead remains manageable even as the dataset size and max $k$ grows, providing superior performance with time efficiency.

which evaluates the quality of a prediction set relative to the true relevant set under a recommendation metric $M \in \{\text{Recall}, \text{MRR}, \text{F1}, \text{NDCG}, \ldots\}$. For a given user $u$, the utility is written as $U_M(I_{\text{true}}(u), \pi_\lambda(u))$, which measures how well the prediction set $\pi_\lambda(u)$ captures the true preferences of user $u$ according to metric $M$.[1]

Subsequently, we define the user-utility-based loss function for user $u$ as follows:

$$L_u(\lambda) = 1 - U_M\big(I_{\text{true}}(u), \pi_\lambda(u)\big). \tag{5}$$

Now given Equation (5), we define the expected risk as:

$$R(\lambda) = \mathbb{E}_{u \sim \mathcal{U}}[L_u(\lambda)]. \tag{6}$$

Building on the principle of Conformal Risk Control (Angelopoulos et al., 2024) and the nesting property in Equation (4), we generalize the approach to utility-based losses tailored for recommender systems and define the optimal threshold $\lambda^*$ as follows:

$$\lambda^\star = \sup\{\lambda \in \Lambda : R(\lambda) \leq \alpha\}. \tag{7}$$

In practice, true data distribution is unknown, we use the empirical risk $\hat{R}_n(\lambda)$ to approximate the expected risk $R(\lambda)$ which is given by:

$$\hat{R}_n(\lambda) = \frac{1}{n} \sum_{i=1}^{n} L_{u_i}(\lambda). \tag{8}$$

To this end, we complete the modeling of the proposed framework. To output the dynamic prediction sets for each user by K-Adapt such that the performance guarantee is met, we design a novel algorithm- K-Adapt based on a greedy strategy to obtain the parameter $\hat{\lambda}$ empirically.

The K-Adapt framework is depicted in Figure 2 and the complete procedure of constructing dynamic prediction sets is summarized in Algorithm 1 in Section A.3 Appendix.

**Prediction Set Construction:** After obtaining $\hat{\lambda}$ from Algorithm 1, we construct prediction sets for $\mathcal{D}_{\text{test}}$. For each user $u$ in the test dataset, we create the prediction set by selecting items with

---

[1]The definitions of the utility metrics are provided in Section A.2.

relevance scores greater than $\hat{\lambda}$. The prediction sets, tailored to individual user preferences, are ensured to control the risk below the user-defined risk threshold $\alpha$.

## 4 THEORETICAL ANALYSIS

In the previous sections, we demonstrated how our framework utilizes a trained model $f_\theta(u, i)$ together with a calibration dataset $\mathcal{D}_{\text{calib}}$ to learn a threshold $\hat{\lambda}$, which is then used to generate dynamic recommendation sets for each user during testing. However it remains to be established whether this empirically calibrated threshold can provide formal guarantees on controlling risk and achieving performance. In this section, we address this by deriving three complementary results. Firstly, we establish that the expected risk under the calibrated threshold is controlled at the user-specified level $\alpha$ up to a finite-sample slack *(validity)*. Secondly, we show that the learned threshold $\hat{\lambda}$ lies close to the population-optimal threshold $\lambda^\star$ *(optimality)*. Finally, we prove that the calibrated threshold is robust to sampling perturbations in the calibration set *(stability)*. The results have been depicted in the following theorems:

**Theorem 1** (Expected Risk Control). *Let $\Lambda$ be a finite set of $\lambda$. The expected risk $R(\lambda)$ is right-continuous in $\lambda$, and is bounded within $[0, B]$ for some $B > 0$ and all $u$ and $\lambda$.*
*For any $\eta > 0$, let $\delta(\eta)$ be the distribution deviation between the expected and empirical risk such that it satisfies:* $2|\Lambda| \exp\left(-\frac{2n\,\delta(\eta)^2}{B^2}\right) \leq \eta$. *Then, for $\hat{\lambda} \in \Lambda$, with probability at least $1 - \eta$ we have,*

$$R(\hat{\lambda}) \leq \alpha + \frac{B}{\sqrt{2n}}\sqrt{\ln\left(\frac{2|\Lambda|}{\eta}\right)}.$$

*Proof.* Proof can be found in Section A.1.1 in *Appendix*. $\qquad\square$

**Remark 1.** *From Theorem 1, we can see the expected risk can be upper bounded by $\alpha$ and a constant term. When the sample size $n \to \infty$, $\frac{B}{\sqrt{2n}}\sqrt{\ln\left(\frac{2|\Lambda|}{\eta}\right)} \to 0$, the upper bound of the expected risk approaches to $\alpha$.*

**Theorem 2** (Optimality of the Calibrated Threshold). *Given $\lambda^\star = \sup\{\lambda \in \Lambda : R(\lambda) \leq \alpha\}$ denote the population-optimal threshold. Let $\hat{\lambda} = \max\{\lambda \in \Lambda : \hat{R}(\lambda) \leq \alpha\}$, where $\hat{R}(\lambda)$ is the empirical risk, suppose $R(\lambda)$ is nondecreasing in $\lambda$ and satisfies the margin condition at $\lambda^\star$ such that there exists $c > 0$ with $R(\lambda) - R(\lambda^\star) \geq c(\lambda - \lambda^\star), \quad \forall \lambda \geq \lambda^\star$. Then with probability at least $1 - \eta$:*

$$-\frac{\varepsilon_n}{c} \leq \lambda^\star - \hat{\lambda} \leq \Delta_\Lambda,$$

*where $\Delta_\Lambda = \max_j(\lambda_{j+1} - \lambda_j)$ and $\varepsilon_n = \frac{B}{\sqrt{2n}}\sqrt{\ln\left(\frac{2|\Lambda|}{\eta}\right)}$ is the deviation term from Theorem 1.*

*Proof.* Proof can be found in Section A.1.2 in *Appendix*. $\qquad\square$

**Remark 2.** *Theorem 2 shows that calibrated threshold $\hat{\lambda}$ produced by K-Adapt is near-optimal as with high probability it lies within one grid step to the left of $\lambda^\star$ and within $\varepsilon_n/c$ to the right of $\lambda^\star$. As the calibration size $n \to \infty$ (so $\varepsilon_n \to 0$) and the grid is refined ($\Delta_\Lambda \to 0$), $\hat{\lambda}$ converges to $\lambda^\star$.*

**Theorem 3** (Stability of the Calibrated Threshold). *Given be a finite set of $\lambda$, and suppose per-user losses satisfy $L_u(\lambda) \in [0, B]$ for all $u, \lambda$. We define the population risk $R(\lambda) = \mathbb{E}[L_u(\lambda)]$, which is nondecreasing in $\lambda$. For a target risk level $\alpha \in (0, 1)$, we define the empirical thresholds as $\hat{\lambda} = \max\{\lambda \in \Lambda : \hat{R}_n(\lambda) \leq \alpha\}, \qquad \hat{\lambda}_{\text{aug}} = \max\{\lambda \in \Lambda : \hat{R}_{n+1}(\lambda) \leq \alpha\}$, where $\hat{R}_n$ and $\hat{R}_{n+1}$ denote the empirical risks computed on $n$ and $n+1$ calibration samples, respectively. We assume the margin condition at the population threshold $\lambda^\star = \sup\{\lambda \in \Lambda : R(\lambda) \leq \alpha\}$ such that there exists $c > 0$ for all $\lambda \geq \lambda^\star$, $R(\lambda) - R(\lambda^\star) \geq c(\lambda - \lambda^\star)$. Then, for any $\eta \in (0, 1)$, with probability at least $1 - \eta$,*

$$|\hat{\lambda}_{\text{aug}} - \hat{\lambda}| \leq \Delta_\Lambda + \frac{2\varepsilon_n}{c},$$

*where $\Delta_\Lambda = \max_j(\lambda_{j+1} - \lambda_j)$ and $\varepsilon_n = \frac{B}{\sqrt{2n}}\sqrt{\ln\left(\frac{2|\Lambda|}{\eta}\right)}$ is the deviation term from Theorem 1.*

*Proof.* Proof can be found in Section A.1.3 in *Appendix*. $\qquad\square$

**Remark 3.** *Theorem 3 shows that the calibrated threshold is robust to perturbations in the calibration set. Its variability decays at rate $O(1/\sqrt{n})$, up to grid resolution $\Delta_\Lambda$. As $n \to \infty$ and $\Delta_\Lambda \to 0$, the thresholds $\hat{\lambda}$ and $\hat{\lambda}_{\text{aug}}$ converge.*

Together, these guarantees demonstrate that K-Adapt not only enforces risk control but also produces thresholds that are provably near-optimal and stable, ensuring both theoretical soundness and practical reliability.

Table 1: Performance comparison between K-Adapt (Ours) and baseline methods under various BaseModels (DeepFM, LighGCN, GMF, MLP, NeuMF), metrics (Recall, MRR, F1, NDCG) across different Datasets (MovieLens, Last.fM, AmazonOffice). For K-Adapt, $\alpha$ and $\eta$ are set empirically as 0.05, respectively. Bold indicates best result and underline marks the second best.

| BaseModel | Method | MovieLens | | | | Last.fM | | | | AmazonOffice | | | |
|---|---|---|---|---|---|---|---|---|---|---|---|---|---|
| | | Recall | MRR | F1 | NDCG | Recall | MRR | F1 | NDCG | Recall | MRR | F1 | NDCG |
| DeepFM | Oracle | 0.47 | 0.72 | 0.37 | 0.47 | 0.47 | 0.70 | 0.38 | 0.46 | 0.52 | 0.32 | 0.23 | 0.28 |
| | Avg-K | 0.36 | 0.59 | 0.23 | 0.29 | 0.33 | 0.50 | 0.25 | 0.31 | 0.32 | 0.19 | 0.11 | 0.20 |
| | AttnCut | 0.38 | 0.62 | 0.29 | 0.32 | 0.32 | 0.58 | 0.31 | 0.35 | 0.33 | 0.14 | 0.15 | 0.16 |
| | MtCut | 0.39 | 0.61 | 0.29 | 0.32 | 0.36 | 0.58 | 0.31 | 0.38 | 0.37 | 0.15 | 0.15 | 0.17 |
| | PerK | 0.41 | 0.62 | 0.30 | 0.37 | 0.40 | 0.61 | 0.32 | 0.40 | 0.44 | 0.23 | 0.17 | 0.21 |
| | K-Adapt (Ours) | **0.43** | **0.67** | **0.33** | **0.42** | **0.43** | **0.65** | **0.34** | **0.43** | **0.48** | **0.28** | **0.18** | **0.23** |
| LightGCN | Oracle | 0.50 | 0.72 | 0.39 | 0.45 | 0.51 | 0.67 | 0.39 | 0.47 | 0.51 | 0.34 | 0.23 | 0.30 |
| | Avg-K | 0.35 | 0.59 | 0.25 | 0.33 | 0.41 | 0.52 | 0.30 | 0.37 | 0.42 | 0.23 | 0.13 | 0.20 |
| | AttnCut | 0.37 | 0.61 | 0.28 | 0.36 | 0.37 | 0.54 | 0.31 | 0.36 | 0.35 | 0.23 | 0.15 | 0.19 |
| | MtCut | 0.41 | 0.64 | 0.29 | 0.36 | 0.39 | 0.56 | 0.32 | 0.38 | 0.37 | 0.25 | 0.15 | 0.21 |
| | PerK | 0.43 | 0.62 | 0.30 | 0.38 | 0.43 | 0.50 | 0.32 | 0.36 | 0.47 | 0.27 | 0.16 | 0.23 |
| | K-Adapt (Ours) | **0.45** | **0.68** | **0.34** | **0.40** | **0.47** | **0.64** | **0.34** | **0.42** | **0.48** | **0.29** | **0.18** | **0.25** |
| GMF | Oracle | 0.41 | 0.67 | 0.32 | 0.38 | 0.46 | 0.61 | 0.37 | 0.45 | 0.47 | 0.28 | 0.21 | 0.28 |
| | Avg-K | 0.17 | 0.53 | 0.21 | 0.20 | 0.41 | 0.55 | 0.31 | 0.38 | 0.41 | 0.21 | 0.13 | 0.22 |
| | AttnCut | 0.29 | 0.58 | 0.25 | 0.31 | 0.26 | 0.52 | 0.32 | 0.35 | 0.31 | 0.20 | 0.12 | 0.19 |
| | MtCut | 0.31 | 0.60 | 0.24 | 0.33 | 0.27 | 0.54 | 0.32 | 0.37 | 0.35 | 0.20 | 0.12 | 0.21 |
| | PerK | 0.35 | 0.57 | 0.25 | 0.32 | 0.40 | 0.55 | 0.32 | 0.38 | 0.44 | 0.18 | 0.13 | 0.21 |
| | K-Adapt (Ours) | **0.38** | **0.62** | **0.27** | **0.34** | **0.42** | **0.57** | **0.34** | **0.41** | **0.46** | **0.24** | **0.16** | **0.24** |
| MLP | Oracle | 0.48 | 0.70 | 0.37 | 0.43 | 0.47 | 0.67 | 0.40 | 0.45 | 0.46 | 0.30 | 0.22 | 0.27 |
| | Avg-K | 0.25 | 0.61 | 0.23 | 0.30 | 0.24 | 0.44 | 0.19 | 0.23 | 0.37 | 0.16 | 0.11 | 0.14 |
| | AttnCut | 0.39 | 0.60 | 0.26 | 0.31 | 0.21 | 0.54 | 0.30 | 0.38 | 0.29 | 0.18 | 0.13 | 0.20 |
| | MtCut | 0.41 | 0.60 | 0.27 | 0.36 | 0.23 | 0.56 | 0.30 | 0.40 | 0.33 | 0.19 | 0.13 | 0.20 |
| | PerK | 0.41 | 0.61 | 0.29 | 0.38 | 0.33 | 0.57 | 0.34 | 0.39 | 0.41 | 0.18 | 0.15 | 0.21 |
| | K-Adapt (Ours) | **0.44** | **0.66** | **0.33** | **0.40** | **0.43** | **0.63** | **0.37** | **0.42** | **0.43** | **0.26** | **0.16** | **0.24** |
| NeuMF | Oracle | 0.51 | 0.74 | 0.39 | 0.47 | 0.50 | 0.71 | 0.40 | 0.49 | 0.50 | 0.31 | 0.23 | 0.30 |
| | Avg-K | 0.38 | 0.61 | 0.24 | 0.33 | 0.31 | 0.48 | 0.19 | 0.25 | 0.32 | 0.22 | 0.12 | 0.22 |
| | AttnCut | 0.40 | 0.63 | 0.25 | 0.34 | 0.35 | 0.55 | 0.32 | 0.39 | 0.34 | 0.22 | 0.15 | 0.20 |
| | MtCut | 0.40 | 0.64 | 0.26 | 0.38 | 0.38 | 0.57 | 0.34 | 0.40 | 0.37 | 0.24 | 0.14 | 0.19 |
| | PerK | 0.43 | 0.62 | 0.32 | 0.39 | 0.42 | 0.58 | **0.36** | 0.42 | 0.44 | 0.23 | 0.16 | **0.25** |
| | K-Adapt (Ours) | **0.46** | **0.69** | **0.34** | **0.42** | **0.45** | **0.67** | **0.36** | **0.44** | **0.46** | **0.25** | **0.18** | **0.25** |

## 5 EXPERIMENTS

In this section, we conduct experiments to validate the effectiveness of the proposed framework (K-Adapt). We design experiments to **1)** validate whether the framework can ensure guaranteed performance by controlling risk below user-defined thresholds and comparing it to other adaptive-k baselines; **2)** analyze the time-efficiency of K-Adapt compared to other baselines. and **3)** analyze how the error rate $\alpha$ and the confidence parameter $\eta$ influences the performance and the average optimal prediction set sizes. **4)** study the stability of K-Adapt as the calibration set size varies

(Section A.5.1); **5)** compare the distribution of calibrated recommendation sizes against oracle and adaptive-$k$ baselines (Section A.5.2); and **6)** examine robustness to user heterogeneity by comparing global and groupwise calibration (Section A.5.3).

## 5.1 DATASETS AND BASELINE METHODS

We experiment on three real-world datasets- MovieLens 100k (Movies) (McAuley et al., 2015), Last.fM (Music) (Cantador et al., 2011) and AmazonOffice (eCommerce) (Harper & Konstan, 2015). To obtain relevance scores, we use five widely recognized recommender models representing diverse architectures: a) DeepFM (Guo et al., 2017); b) LightGCN He et al. (2020); c) GMF (Koren et al., 2009); d) MLP (Zhang et al., 2019) and e) NeuMF He et al. (2017). To evaluate the effectiveness of K-Adapt, we compare it against the following baseline models:

- **AttnCut(Wu et al., 2022):** Employs a Bi-LSTM and Transformer encoder in a classification framework to predict the optimal cutoff position in ranked lists.
- **MtCut (Wang et al., 2022):** Enhances AttnCut using the Multi-gate Mixture-of-Experts (MMoE) model, leveraging multi-task learning for improved cutoff prediction.
- **PerK (KWEON et al., 2024):** Utilizes Poisson-Binomial approximation to compute the expected utility at each cutoff position in ranked lists.

Additionally, we introduce the Avg-K method, where we calculate the average of the prediction set sizes returned by K-Adapt during calibration and use it as a fixed $k$ value for all users. Full implementation details are provided in Section A.4 in Appendix.

## 5.2 EXPERIMENTAL RESULTS

We evaluate the performance of all methods, i.e., Avg-K, AttnCut, Mt-Cut, PerK, and K-Adapt, in terms of HR, Recall, NDCG, and MRR across three datasets: MovieLens, LastFM, and AmazonOffice, implemented on five base recommendation models: DeepFM, LightGCN, GMF, MLP, and NeuMF. The detailed re-

Table 2: Average time (sec) on various datasets.

| Method | Movielens | Last.fM | Amazon Office |
|---|---|---|---|
| AttnCut | 125.67 | 601.67 | 905.88 |
| MtCut | 425.13 | 724.08 | 1017.15 |
| PerK | 1205.78 | 3905.78 | 7560.67 |
| **K-Adapt (Ours)** | **24.08** | **55.27** | **94.28** |

sults are presented in Table 1. From these results, we make the following observations:

- The proposed K-Adapt framework efficiently controls the risk within $\alpha = 0.05$ compared to Oracle values for all tested datasets and metrics across all baselines, thereby aligning with the theoretical expectations. In doing so, it also demonstrates superior performance compared to other baselines.
- The Avg-K method, which uses a fixed set size based on the average prediction sizes of K-Adapt, serves as a competitive baseline. However, its performance declines compared to adaptive methods, particularly on dense datasets like MovieLens, underscoring the importance of personalized prediction sizes for optimal user satisfaction.
- Adaptive methods like AttnCut and MtCut generally outperform Avg-K on dense datasets like MovieLens. However, their performance deteriorates on sparser datasets like Last.fM and AmazonOffice due to reliance on high-dimensional features (e.g., embeddings), which leads to overfitting and poorer generalization in sparse data environments.
- PerK outperforms AttnCut and MtCut by leveraging calibrated interaction probabilities and modeling of user-specific interaction likelihoods using Bernoulli-Poisson framework. This enables it to generalize effectively across both dense and sparse datasets, outperforming methods less suited to sparse or noisy environments. However, it still performs inferior to K-Adapt because of its reliance on determining calibrated interaction probabilities, which, despite user-wise calibration, may fail to fully adapt to user preference variability in highly dynamic environments.
- Additionally, the choice of the base recommendation model significantly impacts the performance of adaptive $k$-based methods. Models like LightGCN and NeuMF consistently outperform GMF and MLP, underscoring the importance of selecting a robust base model to maximize the effectiveness of adaptive frameworks.
- Overall, the results demonstrate K-Adapt's data- and model-agnostic nature, achieving superior performance across all metrics, base models, and datasets by dynamically adjusting recommendation set sizes using Conformal Risk Control.

### 5.3 SPACE AND TIME ANALYSIS

We analyze the computational cost(training time) of the Top-Adaptive-K framework in comparison with other dynamic-K methods. The results averaged on top of all BaseModels are presented in Table 2. From the results, we can observe that our proposed framework is significantly more time-efficient than the other dynamic-K baselines, which indicates the scalability of our method. This is because methods like AttnCut rely on neural models like Bi-LSTM ($h$ (hidden layers)) and Transformer Encoder ($d$) (embeddings), making it computationally heavy with complexities dependent on $h^2$ and $d^2$, leading to time complexity of $\mathcal{O}(u \cdot n \cdot (h^2 + d^2))$, where n is prediction set size. MtCut extends this further by introducing multiple experts ($e$), significantly increasing the resource demands due to the additional model parameters ($e \cdot d^2$), which results in time complexity of $\mathcal{O}(u \cdot n \cdot (h^2 + e \cdot d^2))$. PerK avoids neural networks, relying instead on runtime utility estimations for each user individually, which makes it less resource-intensive during calibration but computationally expensive during inference, particularly when the prediction set size ($n$) or the range of $k$ i.e., $m$) is large. Our framework alleviates these issues by not using neural models and learning $\lambda$ during calibration, avoiding the need to optimize across the range of $k$ (up to $m$) at runtime. As a result, it has efficient time complexity ($\mathcal{O}(u \cdot n \log n)$) that is independent of $m$, unlike other frameworks. This makes K-Adapt highly resource-efficient in practical applications.

### 5.4 PARAMETER ANALYSIS

We further analyze the influence of parameters $\alpha$ (risk threshold) and $\eta$ (confidence threshold) on the performance of the K-Adapt framework. Specifically, we evaluate their impact on performance metrics (e.g., NDCG, F1 score etc.) and average prediction set size.

Figure 4 reports the impact of risk threshold $\alpha$ varying from 0.10 to 0.50 (in increments of 0.05) on average prediction set sizes under fixed confidence threshold $\eta = 0.10$ using Last.fM dataset on Recall and MRR metrics respectively. We observe that as $\alpha$ increases, both average prediction set size and performance metrics exhibit a decreasing trend. This behavior aligns with theoretical expectation, as increasing $\alpha$ relaxes the risk threshold, allowing the model to generate smaller prediction sets but at lower performance. Figure 5 evaluates the impact of confidence level $\eta$ varying from 0.10 to 0.50 (in increments of 0.05) on average prediction set sizes under fixed risk threshold $\alpha = 0.10$ using the AmazonOffice dataset on F1 and NDCG metrics, respectively. We observe that when $\eta$ increases, the model becomes less conservative, leading to a reduction in both prediction set size and performance metrics. This phenomenon demonstrates the framework's ability to balance between prediction set tightness and performance guarantees based on confidence threshold.

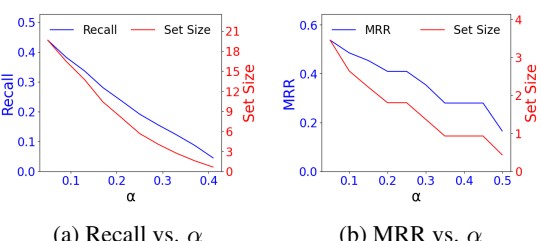

(a) Recall vs. $\alpha$      (b) MRR vs. $\alpha$

Figure 4: Performance trends on the **Last.fm** dataset with varying $\alpha$ and fixed $\eta = 0.1$.

This analysis offers valuable insights into prediction control, enabling practitioners to dynamically adjust the prediction set size and associated performance metrics based on desired risk and confidence thresholds. Due to space constraints, the remaining plots showing similar trends are in the code repository.

### 6 CONCLUSION

This paper introduces limitations of fixed prediction set sizes in RS, which cause user dissatisfaction, degrading RS performance. We propose K-Adapt, a framework that dynamically outputs prediction set sizes while also providing theoretical performance guarantees. Empirical results validate that K-Adapt outperforms heuristic dynamic-$k$ baselines, achieving superior performance with empirically well-chosen risk threshold ($\alpha$) and confidence levels ($\eta$). This work establishes foundations for dynamic, personalized prediction sets with guaranteed performance in diverse recommendation settings.

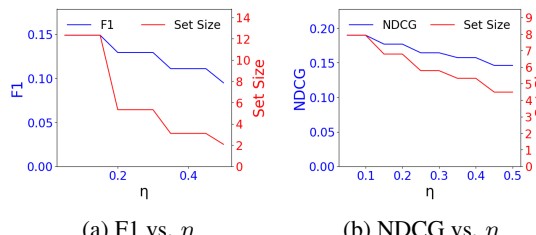

(a) F1 vs. $\eta$      (b) NDCG vs. $\eta$

Figure 5: Performance trends on the **AmazonOffice** dataset with varying $\eta$ and fixed $\alpha = 0.1$.

## 7 ETHICS STATEMENT

All datasets used in the work are publicly available, and the code repository is anonymized; no personal identifying information is involved. The study was conducted in accordance with guidelines for responsible research and reproducible science.

## 8 REPRODUCIBILITY STATEMENT

To facilitate reproducibility, we provide the following resources. 1) Source code and datasets: An anonymized implementation of our proposed framework, supporting codes and datasets are included in the anonymous repository https://anonymous.4open.science/r/Top-Adaptive-K- 551B 2)Proofs: Formal statements and complete proofs underpinning our framework are provided in Section A.1 in the Appendix. 3) Hyperparameters and Implementation Details: The detailed implementation details and configurations are present in Section A.4 in the Appendix.

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

# A APPENDIX

## A.1 PROOFS

### A.1.1 THEOREM 1

*Proof.* We first consider the empirical risk on $n + 1$ samples and using eq. (8):

$$\hat{R}_{n+1}(\lambda) = \frac{1}{n+1} \sum_{i=1}^{n+1} L_{u_i}(\lambda)$$

$$= \frac{n}{n+1} \hat{R}(\lambda) + \frac{1}{n+1} L_{u_{n+1}}(\lambda). \tag{9}$$

where $\hat{R}(\lambda)$ is empirical risk based on the first $n$ samples, and $L_{u_{n+1}}(\lambda)$ is loss on the $(n+1)$th sample.

For $\epsilon > 0$, suppose we have:

$$\hat{R}(\hat{\lambda}) \leq \alpha - \epsilon. \tag{10}$$

Then from equation 9 we get

$$\hat{R}_{n+1}(\hat{\lambda}) \leq \frac{n}{n+1} (\alpha - \epsilon) + \frac{1}{n+1} L_{u_{n+1}}(\hat{\lambda}).$$

Multiplying both sides by $(n + 1)$ and rearranging:

$$(n+1) \hat{R}_{n+1}(\hat{\lambda}) \leq n(\alpha - \epsilon) + L_{u_{n+1}}(\hat{\lambda}),$$

$$L_{u_{n+1}}(\hat{\lambda}) \geq (n+1) \hat{R}_{n+1}(\hat{\lambda}) - n(\alpha - \epsilon).$$

Taking expectations on both sides results in:

$$\mathbb{E}\big[L_{u_{n+1}}(\hat{\lambda})\big] \geq (n+1) \mathbb{E}\big[\hat{R}_{n+1}(\hat{\lambda})\big] - n(\alpha - \epsilon).$$

Since $\mathbb{E}[L_{u_{n+1}}(\lambda)] = R(\lambda)$, we have

$$R(\hat{\lambda}) \geq (n+1) \mathbb{E}\big[\hat{R}_{n+1}(\hat{\lambda})\big] - n(\alpha - \epsilon).$$

Next, we relate $\hat{R}(\lambda)$ to $R(\lambda)$ through Hoeffding's inequality. For any fixed $\lambda$, Hoeffding's inequality states:

$$P\Big( \big|R(\lambda) - \hat{R}(\lambda)\big| > \delta \Big) \leq 2 \exp\Big(-\tfrac{2n\,\delta^2}{B^2}\Big), \tag{11}$$

where $n$ is the number of samples, $\delta$ is deviation between expected risk and empirical risk and $B$ is the bound of risk. Since $\Lambda$ is a finite set of thresholds, we apply union bound over all $\lambda \in \Lambda$. This results in:

$$P\Big( \exists\, \lambda \in \Lambda : \big|R(\lambda) - \hat{R}(\lambda)\big| > \delta \Big) \leq$$

$$\sum_{\lambda \in \Lambda} P\Big(\big|R(\lambda) - \hat{R}(\lambda)\big| > \delta \Big) \leq 2|\Lambda| \exp\Big(-\frac{2n\,\delta^2}{B^2}\Big). \tag{12}$$

Hence,

$$P\Big( \sup_{\lambda \in \Lambda} \big|R(\lambda) - \hat{R}(\lambda)\big| > \delta \Big) \leq 2|\Lambda| \exp\Big(-\tfrac{2n\,\delta^2}{B^2}\Big),$$

and equivalently,

$$P\Big( \sup_{\lambda \in \Lambda} \big|R(\lambda) - \hat{R}(\lambda)\big| \leq \delta \Big) \geq 1 - 2|\Lambda| \exp\Big(-\tfrac{2n\,\delta^2}{B^2}\Big).$$

We now choose $\delta = \delta(\eta)$ to ensure this event has probability at least $1 - \eta$. Concretely, we set

$$2|\Lambda| \, \exp\!\Big(-\tfrac{2n \, \delta(\eta)^2}{B^2}\Big) \;=\; \eta,$$

implying that with probability at least $1 - \eta$,

$$\sup_{\lambda \in \Lambda} \big| R(\lambda) - \hat{R}(\lambda) \big| \;\leq\; \delta(\eta).$$

In particular, for any specific $\lambda \in \Lambda$:

$$|R(\lambda) - \hat{R}(\lambda)| \;\leq\; \delta(\eta).$$

Let us return to $\hat{\lambda}$. From $\hat{R}(\hat{\lambda}) \leq \alpha - \epsilon$ in equation 10 and the bound $|R(\hat{\lambda}) - \hat{R}(\hat{\lambda})| \leq \delta(\eta)$, we obtain

$$R(\hat{\lambda}) \;\leq\; \hat{R}(\hat{\lambda}) + \delta(\eta) \;\leq\; (\alpha - \epsilon) + \delta(\eta).$$

Since $\epsilon$ can be arbitrarily small, we typically write

$$R(\hat{\lambda}) \;\leq\; \alpha + \delta(\eta).$$

Finally, plugging

$$\delta(\eta) \;=\; \frac{B}{\sqrt{2n}} \sqrt{\ln\!\Big(\tfrac{2|\Lambda|}{\eta}\Big)}$$

into the above with probability at least $1 - \eta$ gives:

$$R(\hat{\lambda}) \;\leq\; \alpha \;+\; \frac{B}{\sqrt{2n}} \sqrt{\ln\!\Big(\tfrac{2|\Lambda|}{\eta}\Big)}$$

Hence Proved. $\qquad\square$

### A.1.2 THEOREM 2

*Proof.* By Hoeffding's inequality and a union bound over the finite grid $\Lambda$, with probability at least $1 - \eta$, we have:

$$\sup_{\lambda \in \Lambda} |R(\lambda) - \hat{R}(\lambda)| \;\leq\; \varepsilon_n. \tag{i}$$

We know the empirical $\hat{\lambda}$ chosen by our algorithm satisfies $\hat{R}(\hat{\lambda}) \leq \alpha$, which implies that:

$$R(\hat{\lambda}) \;\leq\; \hat{R}(\hat{\lambda}) + \varepsilon_n \;\leq\; \alpha + \varepsilon_n. \tag{ii}$$

Meanwhile, for any $\lambda > \hat{\lambda}$, the maximality of $\hat{\lambda}$ ensures $\hat{R}(\lambda) > \alpha$, hence

$$R(\lambda) \;\geq\; \hat{R}(\lambda) - \varepsilon_n \;>\; \alpha - \varepsilon_n. \tag{iii}$$

From (ii) and (iii), we get that $\hat{\lambda}$ lies between the population error levels $\alpha - \varepsilon_n$ and $\alpha + \varepsilon_n$. Now we compare $\hat{\lambda}$ with $\lambda^\star$. If $\hat{\lambda} \geq \lambda^\star$, then by the margin condition at $\lambda^\star$ we have:

$$c(\hat{\lambda} - \lambda^\star) \;\leq\; R(\hat{\lambda}) - R(\lambda^\star) \;\leq\; (\alpha + \varepsilon_n) - \alpha \;=\; \varepsilon_n,$$

which gives us

$$\hat{\lambda} - \lambda^\star \;\leq\; \tfrac{\varepsilon_n}{c}, \quad \text{i.e.} \quad \lambda^\star - \hat{\lambda} \leq -\tfrac{\varepsilon_n}{c}. \tag{iv}$$

On the contrary, if $\hat{\lambda} \leq \lambda^\star$, then consider the consecutive grid points $\lambda_j < \lambda^\star < \lambda_{j+1}$. By monotonicity,

$$R(\lambda_j) \;\leq\; \alpha \;\leq\; R(\lambda_{j+1}).$$

This condition ensures $\lambda_j \leq \hat{\lambda} \leq \lambda^\star < \lambda_{j+1}$, such that

$$0 \;\leq\; \lambda^\star - \hat{\lambda} \;\leq\; \lambda_{j+1} - \lambda_j \;\leq\; \Delta_\Lambda. \tag{v}$$

From (iv) and (v), we get that with probability at least $1 - \eta$,

$$-\tfrac{\varepsilon_n}{c} \;\leq\; \lambda^\star - \hat{\lambda} \;\leq\; \Delta_\Lambda.$$

Hence Proved. $\qquad\square$

### A.1.3  THEOREM 3

*Proof.* By Hoeffding's inequality, for each sample size $m \in \{n, n+1\}$ and any $\delta > 0$,

$$\mathbb{P}\Big(\sup_{\lambda \in \Lambda} |R(\lambda) - \hat{R}_m(\lambda)| > \varepsilon_m\Big) \; \leq \; 2|\Lambda| \, \exp\Big(-\frac{2m\varepsilon_m^2}{B^2}\Big).$$

Choosing $\varepsilon_m = \frac{B}{\sqrt{2m}}\sqrt{\ln\big(\frac{4|\Lambda|}{\eta}\big)}$ ensures that, with probability at least $1 - \eta$, the joint event holds:

$$\sup_{\lambda \in \Lambda} |R(\lambda) - \hat{R}_n(\lambda)| \leq \varepsilon_n, \qquad \sup_{\lambda \in \Lambda} |R(\lambda) - \hat{R}_{n+1}(\lambda)| \leq \varepsilon_{n+1}. \tag{i}$$

By construction of $\hat{\lambda}$, we have $\hat{R}_n(\hat{\lambda}) \leq \alpha$, which implies:

$$R(\hat{\lambda}) \; \leq \; \hat{R}_n(\hat{\lambda}) + \varepsilon_n \; \leq \; \alpha + \varepsilon_n. \tag{ii}$$

Similarly, maximality ensures that for any $\lambda > \hat{\lambda}$, $\hat{R}_n(\lambda) > \alpha$, so:

$$R(\lambda) \; \geq \; \hat{R}_n(\lambda) - \varepsilon_n \; > \; \alpha - \varepsilon_n. \tag{iii}$$

Thus $\hat{\lambda}$ is bracketed between the population error rates at levels $\alpha - \varepsilon_n$ and $\alpha + \varepsilon_n$.
Repeating the same argument with $\hat{R}_{n+1}$ yields the analogous bracket for $\hat{\lambda}_{\text{aug}}$.
By monotonicity of $R$ and the margin condition at $\lambda^\star$, any increase of the risk level by $\delta$ can move the cutoff rightward by at most $\delta/c$. Specifically:

$$\sup\{\lambda : R(\lambda) \leq \alpha + \delta\} \; \leq \; \lambda^\star + \delta/c.$$

Applying this with $\delta \in \{\varepsilon_n, \varepsilon_{n+1}\}$ gives

$$\hat{\lambda} \leq \lambda^\star + \varepsilon_n/c, \qquad \hat{\lambda}_{\text{aug}} \leq \lambda^\star + \varepsilon_{n+1}/c. \tag{iii}$$

Conversely, moving the risk level down by $\delta$ can shift the cutoff leftward by at most $\delta/c$, and discretization introduces at most one additional grid step $\Delta_\Lambda$. Hence, we get:

$$\lambda^\star - \varepsilon_n/c - \Delta_\Lambda \; \leq \; \hat{\lambda}, \quad \lambda^\star - \varepsilon_{n+1}/c - \Delta_\Lambda \; \leq \; \hat{\lambda}_{\text{aug}}. \tag{iv}$$

Taking the rightmost admissible value of $\hat{\lambda}_{\text{aug}}$ and the leftmost admissible value of $\hat{\lambda}$ gives:

$$\hat{\lambda}_{\text{aug}} - \hat{\lambda} \; \leq \; \Delta_\Lambda + \frac{\varepsilon_n + \varepsilon_{n+1}}{c}.$$

Finally, since $\varepsilon_{n+1} \leq \varepsilon_n \sqrt{\frac{n}{n+1}}$, we can simplify to

$$|\hat{\lambda}_{\text{aug}} - \hat{\lambda}| \; \leq \; \Delta_\Lambda + \frac{2\varepsilon_n}{c}.$$

Hence Proved. $\qquad\square$

### A.2  UTILITY FUNCTIONS

The design of our loss function in Equation (5) makes the proposed framework more flexible by accommodating different types of utility functions.
Below, we describe the utility functions for commonly used metrics:
1. Utility for Recall:

$$U_{\text{recall}}(I_{\text{true}}(u), \pi_\lambda(u)) = \frac{|I_{\text{true}}(u) \cap \pi_\lambda(u)|}{|I_{\text{true}}(u)|}. \tag{13}$$

This utility measures the proportion of relevant items included in the prediction set. Here, $|\cdot|$ denotes the cardinality of a set.
2. Utility for Mean Reciprocal Rank:

$$U_{\text{mrr}}(I_{\text{true}}(u), \pi_\lambda(u)) = \begin{cases} \dfrac{1}{\min\big\{r(i) \,\big|\, i \in I_{\text{true}}(u) \cap \pi_\lambda(u)\big\}}, \\ \qquad \text{if } I_{\text{true}}(u) \cap \pi_\lambda(u) \neq \emptyset, \\ 0, \quad \text{otherwise.} \end{cases} \tag{14}$$

This utility measures how early the first relevant item appears in the ranked list. The term $r(i)$ represents the rank of item $i$ in the prediction set $\pi_\lambda(u)$.

3. Utility for F1-Score :

$$U_{\text{F1}}(I_{\text{true}}(u), \pi_\lambda(u)) = \frac{2|I_{\text{true}}(u) \cap \pi_\lambda(u)|}{|I_{\text{true}}(u)| + |\pi_\lambda^{\max}(u)|}. \tag{15}$$

This utility balances how many of the relevant items are actually predicted with how many of the predicted items are truly relevant. Here, $|\pi_\lambda^{\max}(u)|$ is the maximum possible size of the prediction set for user $u$ at the given threshold $\lambda$.

4. Utility for NDCG :

$$U_{\text{ndcg}}(I_{\text{true}}(u), \pi_\lambda(u)) = \frac{\sum_{i=1}^{|\pi_\lambda(u)|} \frac{\mathbb{I}[i \in I_{\text{true}}(u)]}{\log_2(i+1)}}{\sum_{i=1}^{|I_{\text{true}}(u)|} \frac{1}{\log_2(i+1)}}. \tag{16}$$

This utility measures ranking quality by assigning higher importance to relevant items appearing earlier in the ranked list. Here, $\mathbb{I}[i \in I_{\text{true}}(u)]$ is an indicator function that returns 1 if the item $i$ is relevant, and 0 otherwise. The term $\log_2(i+1)$ is a position-based discount factor to penalize items ranked lower.

By leveraging utility functions tailored to specific metrics, K-Adapt accommodates a wide range of evaluation criteria such as Recall, MRR, F1 and NDCG.

## A.3 ALGORITHM

We now present the algorithm for the *K-Adapt* framework. The procedure begins with an initialization step where the control parameter $\lambda$ is set to the initial value $\lambda_{\text{init}}$. For each candidate $\lambda$, the algorithm constructs calibrated prediction sets $\pi_\lambda(u)$ (Eq. 4) from the base recommender scores $f_\theta(u, i)$ by retaining all items with scores above the threshold. Next, the user-utility loss $L_u(\lambda)$ is computed for each user as defined in Eq. 5, and aggregated into the empirical risk $\hat{R}_n(\lambda)$ (Eq. 8). To find the largest $\lambda$ such that the empirical risk remains below the target risk level $(\alpha - \epsilon)$, where $\alpha$ is the user-defined tolerance and $\epsilon$ is a slack parameter, we search in a greedy manner. When $\hat{R}_n(\lambda) \leq \alpha - \epsilon$, the algorithm terminates and returns $\hat{\lambda} = \lambda$. Otherwise, the threshold is reduced in steps of $\Delta\lambda^\dagger$ until the stopping condition is satisfied. This simple yet effective strategy ensures that the resulting calibrated threshold $\hat{\lambda}$ balances compactness of recommendation sets with statistical risk control.

The detailed procedure is summarized in Section A.3.

---

**Algorithm 1** K-Adapt Algorithm

---

**Input:** Recommendation model $f_\theta(u, i)$, calibration dataset $\mathcal{D}_{\text{calib}}$, initial control parameter $\lambda_{\text{init}}$, user-defined risk threshold $\alpha$, error tolerance $\epsilon$.
**Output:** Calibrated threshold $\hat{\lambda}$.

1: Define utility function $U_M(\cdot)$ based on the chosen recommendation metric.
2: Initialize $\lambda \leftarrow \lambda_{\text{init}}$.
3: **while** $\lambda > 0$ **do**
4:     Generate prediction set $\pi_\lambda(u)$ using $f_\theta(u, i) \geq \lambda$ ( Equation 4).
5:     Compute user loss $L_u(\lambda)$ ( Equation 5).
6:     Compute empirical risk $\hat{R}_n(\lambda)$ ( Equation 8).
7:     **if** $\hat{R}_n(\lambda) \leq \alpha - \epsilon$ **then**
8:         **return** $\hat{\lambda} \leftarrow \lambda$
9:     **else**
10:        Update $\lambda \leftarrow \lambda - \Delta\lambda^\dagger$.
11:     **end if**
12: **end while**

---

## A.4 IMPLEMENTATION DETAILS

All the base recommender models are trained using the Adam Optimizer. We train each model for 20 epochs while keeping the learning rate at 0.001 and batch size at 256. The scores $f_\theta(u, i)$ generated by these base recommender models are used as inputs for the adaptive k methods. Detailed configurations of the baselines are as follows: MMOECut uses 3 experts with Transformer layers of size 128 and 2 attention heads, combined with a gating mechanism and a bi-directional LSTM of

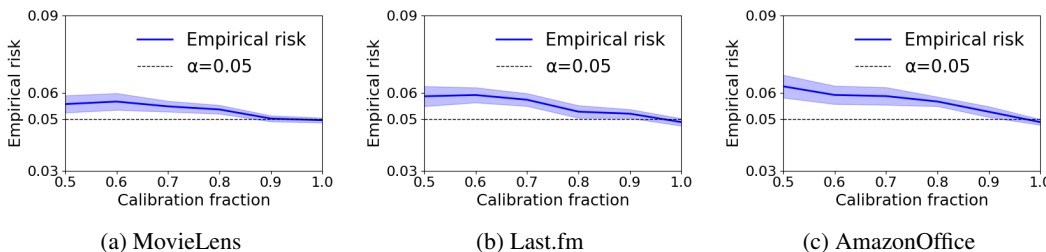

Figure 6: Calibration stability analysis across MovieLens, Last.fM and AmazonOffice datasets respectively. The y-axis reports empirical risk (defined w.r.t. Recall), and the x-axis shows the calibration fraction. The dashed line marks the target $\alpha = 0.05$. Shaded regions indicate 95% confidence intervals across 20 random calibration subsamples.

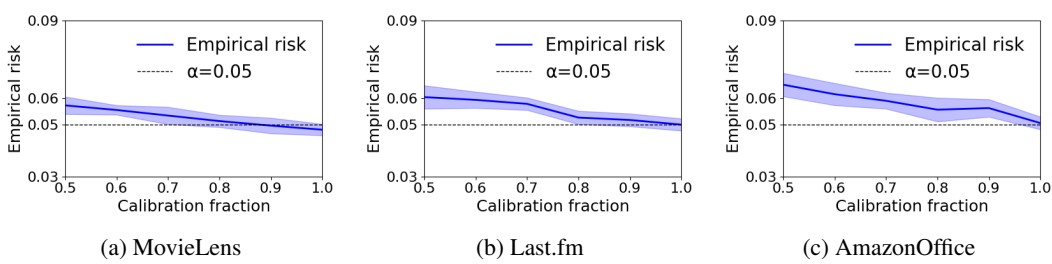

Figure 7: Calibration stability analysis across MovieLens, Last.fM and AmazonOffice datasets respectively. The y-axis reports empirical risk (defined w.r.t. NDCG), and the x-axis shows the calibration fraction. The dashed line marks the target $\alpha = 0.05$. Shaded regions indicate 95% confidence intervals across 20 random calibration subsamples.

size 64; AttnCut employs a Transformer layer of size 64 with 2 attention heads and a bi-directional LSTM of size 32; PerK employs a Poisson-Binomial approximation to compute expected utility values. Our framework K-Adapt uses error rate $\alpha = 0.05$ and confidence parameter $\eta = 0.05$. Furthermore, we split the held-out training data with 60% as the calibration data and 40% as the testing data on all datasets. We also set the negative sampling rate to 50 for each true user-item interaction and the maximum recommendation size for each user to 25.

## A.5 ADDITIONAL EXPERIMENTS

### A.5.1 CALIBRATION SIZE ANALYSIS

We analyze the stability of K-Adapt as the calibration set size varies. We fix the error rate at $\alpha = 0.05$, the confidence level at $\eta = 0.05$, and the maximum recommendation size at $K_{\max} = 25$. We subsample the calibration users from 50% to 100% in 10% increments, recalibrate $\hat{\lambda}$ twenty times per fraction, and evaluate the Recall and NDCG based empirical risk on a fixed test set in Figures 6 and 7 respectively. These plots depict empirical risk (solid line) against the calibration fraction and the shaded band indicates the variability across the twenty recalibrations (95% confidence interval). With only 50% of the calibration data, the achieved risk is close to but slightly above the target $\alpha = 0.05$, and the shaded band is comparatively wide. As the calibration fraction increases, both quantities improve: the mean empirical risk decreases and aligns tightly with $\alpha$, and the shaded band contracts, indicating lower variability. This behavior is expected as the smaller calibration sets result in noisier estimates of $\hat{\lambda}$ and thus more dispersion in achieved risk, whereas larger calibration sets stabilize the estimate and concentrate the risk near the target level. Interestingly, the variance across calibration subsamples also differs by metric as Recall exhibits the tighter band, while NDCG is more volatile due to its sensitivity to single-item rank positions or sparse positives. Overall, K-Adapt maintains risk control even with limited calibration data, and additional calibration users further tighten the guarantees while reducing variance.

---

† $Here, \Delta(\lambda)$ is equivalent to $\frac{\lambda}{|\Lambda|}$, where $\Lambda$ is set of $\lambda$.

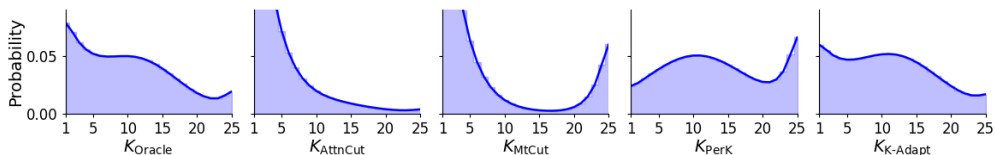

Figure 8: Distribution of recommendation set sizes on **MovieLens** datasets for $K_{\max} = 25, \alpha = 0.05\, \eta = 0.05$. The x-axis denotes the selected recommendation size $K$, and the y-axis shows the empirical probability across users. Results are shown for Oracle, AttnCut, MtCut, PerK, and K-Adapt.

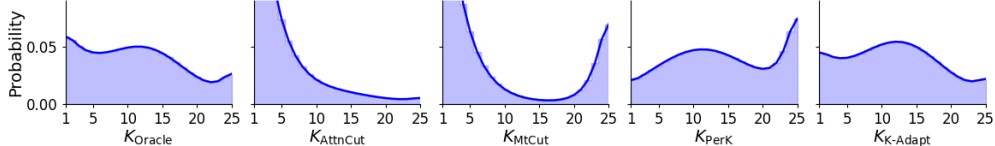

Figure 9: Distribution of calibrated set sizes on **Last.fM** datasets for $K_{\max} = 25, \alpha = 0.05\, \eta = 0.05$. The x-axis denotes the selected recommendation size $K$, and the y-axis shows the empirical probability across users. Results are shown for Oracle, AttnCut, MtCut, PerK, and K-Adapt.

### A.5.2 RECOMMENDATION SIZE DISTRIBUTION ANALYSIS

To further understand the behavior of K-Adapt, we compare the distribution of calibrated recommendation set sizes against the Oracle and adaptive-$k$ baselines. We fix the maximum recommendation size to $K_{\max} = 25$ and evaluate on MovieLens, Last.fm, and AmazonOffice, using NDCG-based utility metric at $\alpha = 0.05$. For each method, we record the empirical distribution of prediction set sizes across all users. The results are presented in Figures 8 to 10. From the firesults we observe:

- Firstly, the oracle distributions differ across datasets: MovieLens produces left-heavy distributions with most users requiring smaller set sizes as it is denser; AmazonOffice, in contrast, is sparse and shifts mass toward larger set sizes; Last.fm lies between these extremes. This highlights the need for user- and dataset-adaptive calibration rather than a one-size-fits-all choice of $K$.

- Among the baselines, AttnCut and MtCut both exhibit strong small-$K$ bias, concentrating probability mass at the left end of the spectrum. MtCut is slightly more spread out due to its expert-gated architecture, but the two remain largely similar, especially in denser datasets.

- The PerK baseline, by design, produces broader mid-range distributions with a visible tail near $K_{\max}$, reflecting its reliance on Poisson–Binomial approximations that are sensitive to calibration noise. Importantly, the strength of this tail grows on sparser datasets, where user-level probabilities are less stable.

- K-Adapt consistently follows the Oracle distributions more closely than the baselines. On MovieLens, it preserves the left-heavy shape while allowing for occasional larger sets; on AmazonOffice, it flexibly shifts mass toward higher $K$ values while avoiding the overextended tails observed in PerK. It, however, has slight deviations which are expected. This deviation is due to K-Adapt, which optimizes risk guarantees without directly observing ground-truth oracle sizes.

Overall, these results show that K-Adapt not only provides theoretical guarantees but also adapts set-size distributions in a way that reflects dataset density and user heterogeneity.

### A.5.3 HETEROGENEITY ANALYSIS BY USER ACTIVITY

We finally examine whether a single global threshold $\hat{\lambda}$ suffices for heterogeneous user populations, or whether group-specific calibration provides noticeable benefits. Following Li et al. (2021) we partition the AmazonOffice dataset into two cohorts based on interaction count: a *low-activity* group and a *high-activity* group. Users are initially split evenly, and the boundary is then adjusted to ensure that the minimum interaction count in the high-activity group exceeds the maximum in the low-activity group by at least one. We then evaluate each group separately on the test set, comparing the performance under the global $\hat{\lambda}$ with that under group-wise thresholds $\hat{\lambda}_g$ calibrated within each cohort. We report Recall, MRR, F1, and NDCG together with the average set size in parentheses. As

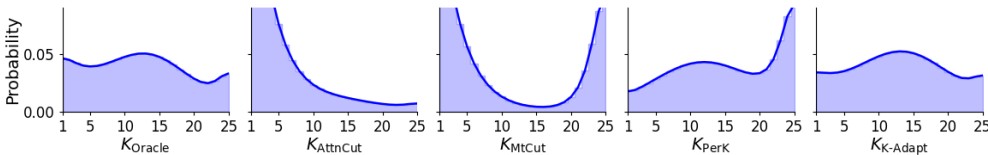

Figure 10: Distribution of calibrated set sizes on **AmazonOffice** datasets for $K_{\max} = 25, \alpha = 0.05\, \eta = 0.05$. The x-axis denotes the selected recommendation size $K$, and the y-axis shows the empirical probability across users. Results are shown for Oracle, AttnCut, MtCut, PerK, and K-Adapt.

shown in Table 3, the results highlight only modest differences between global and group-wise calibration. For low-activity users, group-wise calibration tends to increase set sizes slightly, improving Recall and F1. For high-activity users, the global $\hat{\lambda}$ is mildly conservative, so group-wise calibration yields small improvements in MRR and NDCG by trimming redundant items. However, in both groups the utility gaps are minor, demonstrating that K-Adapt's global calibration already delivers near-oracle performance while providing robust guarantees across heterogeneous user cohorts.

Table 3: Heterogeneity analysis on **AmazonOffice** by user activity ($\alpha = 0.05$, $\eta = 0.05$, $K_{\max} = 25$) using NeuMF. Values in parentheses denote average set size. $\hat{\lambda}$ is the global threshold, while $\hat{\lambda}_{g_1}$ and $\hat{\lambda}_{g_2}$ are group-specific thresholds for low- and high-activity cohorts.

| Group | Calibration | Recall ↑ | MRR ↑ | F1 ↑ | NDCG ↑ |
|---|---|---|---|---|---|
| Low-activity | $\hat{\lambda}$ | 0.429 (16.9) | 0.225 (10.6) | 0.165 (11.4) | 0.234 (12.5) |
| | $\hat{\lambda}_{g_1}$ | 0.442 (17.8) | 0.231 (11.4) | 0.172 (12.6) | 0.248 (13.6) |
| High-activity | $\hat{\lambda}$ | 0.465 (7.6) | 0.255 (6.5) | 0.185 (7.1) | 0.256 (8.5) |
| | $\hat{\lambda}_{g_2}$ | 0.472 (8.05) | 0.264 (7.2) | 0.189 (7.8) | 0.266 (9.0) |
| Gap ($\Delta$) | $\hat{\lambda}$ | 0.036 | 0.030 | 0.020 | 0.022 |
| | $\hat{\lambda}_g$ | 0.030 | 0.033 | 0.017 | 0.018 |

## A.6 DISCUSSION

K-Adapt provides a novel solution to the practical problem of dynamic recommendation set selection: how to adapt the cutoff $K$ flexibly across users while retaining guarantees on recommendation quality. Existing adaptive-$K$ methods either rely on learned heuristics or distributional approximations, which may perform well in specific settings but lack formal guarantees. By converting this problem into a risk-controlled prediction set problem, K-Adapt bridges this gap, offering a data- and model-agnostic framework that balances flexibility in set size with rigorous statistical guarantees ensuring sustained control of utility risk across diverse datasets and base recommenders. Specifically, K-Adapt operates as a post-hoc calibration layer designed for the user-facing truncation stage, effectively resolving the critical trade-off between candidate coverage and display constraints.

Empirically, K-Adapt consistently outperforms baseline adaptive-$K$ methods across MovieLens, Last.fm, and AmazonOffice while preserving the desired risk control. It does so without introducing significant runtime overhead, as calibration is a single forward pass per threshold and scales efficiently with dataset size. Furthermore, K-Adapt is robust to heterogeneous user populations: experiments on AmazonOffice show that both global and group-wise calibration deliver near-oracle utilities, with only minor differences across cohorts. This suggests that one global $\hat{\lambda}$ already provides strong coverage while group-specific calibration can offer additional refinements if needed. K-Adapt is also flexible. Since the utility function $U_M$ is externally defined, the framework can be adapted to optimize for diverse objectives such as diversity, fairness, or safety, while retaining the same statistical guarantees. For instance, $U_M$ could penalize concentration on a few popular items, or enforce constraints on exposure across groups. Validity guarantees would then hold with respect to the modified utility function, requiring no change in the core theory.

At the same time, K-Adapt faces two key challenges. First, as with other conformal methods, finite-sample effects can lead to conservative prediction sets when calibration data are scarce, as observed in our stability analysis. This limitation, however, highlights a natural application to recent Top-$K$ optimization objectives (Yang et al., 2025; Shi et al., 2024; Qiu et al., 2022; Jagerman et al., 2022),

which focus on enhancing the underlying ranking quality. Combining these upstream improvements with K-Adapt's downstream risk control represents a promising direction for future research. Second, K-Adapt depends on the quality of the underlying recommender scores: if a base model produces poorly ranked or miscalibrated scores, the resulting thresholds cannot fully compensate. Addressing these challenges, for example, by integrating improved calibration of base scores or hybrid group-wise updates, remains promising future work. Overall, K-Adapt advances the reliability of adaptive-$K$ recommendation by combining the simplicity of threshold-based cutoff selection with the rigor of conformal risk control. We believe this pragmatic step can inspire future research on trustworthy, utility-aware recommendation systems.

