# OpenReview forum: "Guaranteed Top-Adaptive-K in Recommendation"
_ICLR.cc/2026/Conference — ICLR 2026 Conference Withdrawn Submission_

### Official Review · Reviewer_TAkm · 2025-10-26

**Soundness:** 2
**Presentation:** 2
**Contribution:** 2
**Rating:** 2
**Confidence:** 4

**Summary:**

This paper proposes K-Adapt, a model-agnostic adaptive framework for recommender systems that automatically chooses recommendation size per user via a calibrated score threshold $\lambda$. It formulates the problem as utility-based risk control, extends conformal prediction to ranking metrics, and uses a greedy calibration algorithm. Theoretical analysis provides theoretical guarantees, near-optimality, and perturbation-stability of the proposed method. Experiments on three datasets and five backbones show improvements over selected dynamic-K baselines.

**Strengths:**

- This paper studies the adaptive recommendation size, which is an important and practical problem in RS.
- Theoretical analysis is provided for the proposed method.
- The experimental results demonstrate the effectiveness of the proposed method.

**Weaknesses:**

- The motivation for the proposed adaptive threshold $\lambda$ is not convincing due to the trivial solutions.
- Some critical assumptions in the theoretical analysis are not reasonable.
- The write-up and mathematical notations can be improved for better clarity.

**Questions:**

**Concerns about Motivation.** The proposed adaptive threshold $\lambda$ for restricting recommendations is not convincingly motivated:
- The definition of the optimal $\lambda^*$ assumes that $R(\lambda)$ is non-decreasing (equivalently $U(\lambda)$ is non-increasing), as also stated in Theorem 2. This holds for some metrics (e.g., Recall, NDCG) but fails for others, such as Precision, original F1-score (not the one defined in Appendix A.2), etc. This limits the applicability of the proposed method in real-world RS, and the authors should provide further discussion on this point.
- Even when the assumption holds, minimizing $\lambda$ trivially minimizes $R(\lambda)$. For instance, $\lambda \to -\infty$ yields maximum Recall and NDCG by recommending all items. However, this trivial solution is meaningless in practice. The proposed greedy algorithm with $K_{\max}$ avoids the "all items" case but not the equally trivial "always recommend $K_{\max}$ items", since $\lambda$ only decreases. The authors should clarify how their method avoids such trivial solutions.

**Concerns about Theoretical Results.** Some critical assumptions in the theoretical results are not convincing, which needs to be addressed:
- As stated above, the monotonicity assumption on $R(\lambda)$ may not hold, which invalidates Theorem 2 and subsequent results.
- Theorems 2 and 3 also assume that there exists a positive constant $c$ such that for any $\lambda \ge \lambda^\star$, $R(\lambda) - R(\lambda^\star) \geq c(\lambda - \lambda^\star)$. This assumption is obviously not true in general for almost all metrics due to their discrete nature and stepwise changes.

**Discussion on Top-$K$ Recommendation Optimization.** Recent literature has extensively studied top-$K$ recommendation optimization, including LambdaLoss@$K$ [R1], SONG@$K$ [R2], LLPAUC [R3], and SL@$K$ [R4]. These methods optimize fixed top-$K$ metrics with a fixed $K$, achieving strong empirical performance and theoretical guarantees. Since the proposed K-Adapt method is model-agnostic and can be applied only at inference time, it would be interesting to see whether it can further improve the performance of these top-$K$ recommendation losses.

**Minor Concerns:**

- Theorem 1: What is the meaning of $\lambda^*$ in the proof? Should it be $\hat\lambda$? In addition, it seems that the deviation $\delta(\epsilon)$ is in fact not dependent on $\epsilon$.

**References:**

- [R1] On Optimizing Top-K Metrics for Neural Ranking Models. SIGIR '22.
- [R2] Large-scale Stochastic Optimization of NDCG Surrogates for Deep Learning with Provable Convergence. ICML '22.
- [R3] Lower-Left Partial AUC: An Effective and Efficient Optimization Metric for Recommendation. WWW '24.
- [R4] Breaking the Top-K Barrier: Advancing Top-K Ranking Metrics Optimization in Recommender Systems. KDD '25.

---

> ### Author Response · Authors · 2025-11-27
> **On Weaknesses**
>
> We thank reviewer for recognizing **practicality** and **effectiveness** of our method.  Below we address the critiques:
>
> ---
>
> ### Q1. The proposed greedy algorithm ... avoids  “all items” case but not equally trivial “always recommend K items”
>
> > We respectfully clarify that this concern reflects a minor misinterpretation of fundamental definition of our optimization objective. Our objective is not to minimize $R(\lambda)$; instead, we constrain it to be below $\alpha$ and maximize $\lambda$. **Our method does not minimize $\lambda$. in fact, it maximizes it.**  As defined in Eq. 7, the optimal threshold is the supremum:  $\lambda^* = \sup \{ \lambda \in \Lambda : R(\lambda) \le \alpha \}$. We seek **largest possible $\lambda$** that still satisfies the risk constraint $\alpha$. Due to the nesting property (Eq. 4), maximizing $\lambda$ is mathematically equivalent to **minimizing prediction set size**.  Had we minimized $\lambda$, we agree that we would recommend all items. However, by finding supremum, K-Adapt *searches for the largest $\lambda$ that still keeps empirical risk below $\alpha$*, which corresponds to  smallest recommendation lists compatible with  risk guarantee.
> >
>
> > Our empirical results validate this effectively in **Figures 8, 9, and 10 (Appendix A.5.2)**. The set sizes are distributed dynamically across $[1, K_{\max}]$ based on user uncertainty, rather than clustering at $K_{\max}$ (which would be  result of trivial solution). This confirms K-Adapt avoids triviality by adapting to individual user needs.
>
> ---
>
> ### Q2. The monotonicity assumption holds for some metrics.. but fails for others... This limits  applicability.
>
> > We acknowledge while monotonicity is a required condition for our current guarantees but we respectfully disagree that this limits practical usability of proposed method. Our method aligns and complements the standard **two-stage architecture** widely adopted in real-world recommenders (e.g., YouTube [5]).  Specifically, real-world systems utilize a two-stage pipeline (Retrieval → Ranking). K-Adapt operates as a **post-hoc calibration layer** specifically for  **retrieval stage**, where the primary objective is to maximize **coverage** (Recall) to ensure no relevant items are missed before the expensive ranking phase [5].  Optimizing for precision at this stage is often counter-productive, as it filters out relevant niche candidates prematurely. High precision is primarily driven by the ranking quality of the backbone model or subsequent re-ranking stage. K-Adapt’s specific role is **ensuring risk-controlled retrieval**.
> >
> > For situations requiring a strict balance, we provided a practical F1-Score formulation in Eq. 15.
>
> ---
>
> ### Q3. Concerns about Theoretical Results: The margin condition... is obviously not true in general due to discrete metrics.
>
> > We respectfully highlight that the margin condition is imposed on the **expected risk**, not on individual user losses. While $L_u(\lambda)$ for a single user is a step function, the expected risk  $R(\lambda) = \mathbb{E}[L_u(\lambda)]$** averages these stepwise losses over the population**.  Under mild regularity assumptions on user preference and score distributions, this aggregation behaves approximately smoothly, allowing a margin-type condition to hold at population level.
> >
>
> > Importantly, this assumption **does not affect** the validity of our main guarantee: Theorem 1 (expected risk control) does *not* rely on the margin condition. It is only used for results on near-optimality and stability.  Thus, core safety guarantee of K-Adapt remains valid even if margin condition is relaxed.
>
> ---
>
> ### Q4. Discussion on Top-K Recommendation Optimization: ... may improve performance of top-K optimization losses.
>
> > We agree that K-Adapt’s model-agnostic nature allows it to be directly plugged in on top of top-K–oriented ranking optimization methods [1,2,3,4].  These objectives focus on improving ranking curve, while K-Adapt works **post hoc to optimize the truncation point**, deciding where to cut the ranked list using a calibrated threshold.  These are **complementary** roles: a stronger ranking curve from a top-K trained model can be **more effectively truncated** by K-Adapt under utility-based risk control.
> >
>
> > Empirically, our experiments across five heterogeneous backbones (DeepFM, LightGCN, etc.) show consistent gains, suggesting that K-Adapt is a **backbone-agnostic enhancer**.  We expect similar benefits when combined with models trained using objectives in [1,2,3,4], and **added** these references to Discussion.
>
> ---
>
> ### Q5. Notational concerns
>
> > Thank you for checking notational issues. We have corrected the typo $\lambda^*$ in revision and updated notation to $\delta(\eta)$ to reflect  deviation bound’s dependency on $\eta$.
>
> ---
>
> ### References
>
> [5] Covington, P., et al. (2016). *Deep neural networks for YouTube recommendations.* RecSys.

---

> > ### Comment · Reviewer_TAkm · 2025-11-27
> > **Response to Authors' Rebuttal**
> >
> > The authors' rebuttal has addressed several of my previous concerns, and accordingly I have updated my rating to 4.
> >
> > That said, I still have substantial reservations regarding Q2. The authors assert that "K‑Adapt operates as a post‑hoc calibration layer specifically for the retrieval stage,” yet this point is not explicitly articulated in the paper. In fact, the proposed method does not appear to be inherently limited to retrieval or reranking tasks. Furthermore, the experimental setting with $K_\max \leq 25$ deviates from conventional retrieval scenarios, where the number of retrieved items typically reaches the hundreds. In addition, my concern regarding the monotonicity assumption --- which is central to the authors’ theoretical analysis --- remains insufficiently addressed.

---

> ### Author Response · Authors · 2025-12-02
>
> We thank the reviewer for raising the rating of our work. We appreciate the opportunity to resolve the remaining concerns:
>
> ---
>
> ### 1. On Scope and Experimental Setting
> >We agree  that our method is generalized and not inherently limited to retrieval; however, its primary value lies in the **user-facing truncation** in the retriever stage. Regarding the deviation from conventional large-scale retrieval ($K=100+$), we set $K_{max}=25$ specifically to target the **"critical design"** of user interaction. Specifically, in unconstrained retrieval (e.g., $K=500$), achieving high recall is algorithmically trivial. The true challenge, and the necessity for our optimization, arises precisely in the constrained setting (e.g., mobile screeens) where the system must make trade-offs between coverage and cognitive load. K-Adapt is specifically designed to **maximize utility within this constrained interface**. We have **revised** the submission in red in Introduction and Discussion A.6 to highlight this.
>
> ---
>
> ### 2. Explanation of Monotonicity
> > To resolve the concern about the centrality of this assumption, we distinguish between **validity** and **efficiency**. As acknowledged earlier, **Theorem 1 (Validity/Risk Control)** relies solely on the exchangeability of calibration scores and the boundedness of the loss (via Hoeffding's inequality). It does **not** explicitly assume monotonicity. Consequently, K-Adapt **guarantees validity** that the realized risk will theoretically remain below $\alpha$ with high probability across all metrics, regardless of monotonicity. We acknowledge that monotonicity is an assumption required for Theorem 2 to prove that the greedy solution is the *globally optimal* (smallest) set. However, even if this assumption fails (e.g., strictly non-monotonic metrics), K-Adapt remains **valid**  but may be conservative (producing slightly larger sets than theoretically possible. Thus, the method's core contribution of **risk control** holds universally.

---

### Official Review · Reviewer_4jwe · 2025-10-27

**Soundness:** 2
**Presentation:** 2
**Contribution:** 2
**Rating:** 6
**Confidence:** 4

**Summary:**

The paper is about controlling the score threshold for recommendations.\
To do this, they propose K-adapt (GUARANTEED ADAPTIVE-K IN RECOMMENDATIONS).\
Specifically, based on the calibration dataset, they set the threshold for ensuring the pre-defined performance.\
Then, they use the threshold for the inference to control the number of recommendations for each user.

**Strengths:**

1. The paper is well formulated and easy to follow.
- The paper has a good structure and good citation format.
- Figures and Tables are well organized and presented.

2. Fast inference time.
- They use the pre-selected threshold for the inference time, which results in faster inference than existing methods.

3. Experiment on real-world datasets demonstrates the superiority of the proposed method.

**Weaknesses:**

1. The method should be described in a more detailed way.
- Algorithm 1 should be included in the main manuscript, not in Appendix.

2. Technical contribution is marginal.
- From my understanding, the method selects the threshold to ensure a certain performance in the calibration set.\
Then, it uses the threshold for the inference.
- Is the global threshold enough for all users?

3. Not guaranteed performance?
- In the manuscript and title, they noted that their method "guarantees" the performance.
- In Table 1, however, $\alpha=0.05$ and the performance is not 0.95.
- Also, in Figure 4, recall is almost zero when $\alpha=0.4$.

**Questions:**

Please refer to Weaknesses.\
Also, what is the purpose of Eq.5? I think both cases are the same.

---

> ### Author Response · Authors · 2025-11-27
> **On Weaknesses**
>
> Thankyou for your positive evaluation.  We appreciate the opportunity to clarify the technical details and concerns as below:
>
> ---
>
> ### Q1. Algorithm 1 should be included in the main manuscript, not in Appendix.
>
> > We originally placed Algorithm 1 to the Appendix due to the strict page limit. In the final revision with additional page, we will move Algorithm 1 and the methodology details into the main text. Thanks for the suggestion.
>
> ---
>
> ### Q2. Is the global threshold enough for all users?
>
> > Yes, we highlight that using global threshold $\lambda$ produces highly personalized set sizes and  does **not** imply non-personalized recommendations. Technically, the recommendation score $f_\theta(u,i)$ generated by the base model is inherently personalized (capturing user-item affinity). The global threshold $\lambda$ learned acts as a minimum confidence value.  For a user with clear preferences (high scores), many items will exceed $\lambda$, resulting in a large $K_u$. Conversely, for a user with uncertain preferences (low scores), few items will exceed $\lambda$, resulting in a small $K_u$.  Thus, a single calibrated $\lambda$ **automatically induces personalized $K_u$ values** adapted to each user's confidence profile.
> >
>
> > **Empirically**, we validate this in two ways:
> > 1. **Personalized Distributions:** We plot the empirical distribution of set sizes across users in **Appendix A.5.2 (Figure 8–10)**. The results show a variance in $K_u$ (spanning $[1, 25]$), proving that the **global threshold successfully adapts to individual user uncertainty**.
> > 2. **Group Robustness (Table 3):** We explicitly tested the global threshold against group-specific thresholds (partitioning by user activity) in **Appendix A.5.3**. The performance gap was on the order of 0.01 (e.g., 0.429 vs 0.442 and 0.465 vs 0.472 for Recall), confirming that a **single global threshold is sufficient even for heterogeneous populations.**
>
> ---
>
> ### Q3. Concerns about Guarantees: “Performance is not 0.95… Recall is almost zero when α is high.”
>
> > We respectfully point out that this concern stems from a misunderstanding of how “risk” is defined in bounded top-$K$ recommendation.
> >
>
> > **Relative Guarantee (Table 1 Analysis):**
> > We highlight that recall (or other RS metric) is not 0.95 (despite $\alpha = 0.05$). Instead, the risk guarantee is bounded by the **Model’s Capacity** (Oracle performance), not theoretical perfection (1.0).
> >
> > - **Oracle:** In Table 1, the maximum possible recall with size $K_{\max} = 25$ (chosen to highlight screen/latency constraints) is ≈0.47 (DeepFM) or ≈0.50 (LightGCN). No method can exceed this.
> > - **K-Adapt’s Guaranteed Performance:** K-Adapt achieves ≈0.43 (DeepFM) and ≈0.45 (LightGCN). The gap to Oracle is 0.04, which is within the risk budget $\alpha=0.05$.
> >
> > Thus, the method **successfully guaranteed performance** relative to the achievable ceiling.
> >
>
> > **Interpretation of Figure 4 (High $\alpha$):**
> > The observation that Recall decreases when $\alpha$ increases is the **correct and expected behavior**.
> >
> > - $\alpha$ represents the **allowable error level**. A higher $\alpha$ means the algorithm may use a higher threshold, producing smaller sets.
> > - Smaller sets naturally yield lower Recall.
> >
> > Therefore, the drop in Recall confirms that **K-Adapt correctly responds to the user’s specified risk tolerance**.
>
> ---
>
> ### Q4. Purpose of Eq. 5
>
> > We originally presented it in piecewise form to handle boundary conditions, ensuring loss is bounded in $[0,1]$ even for edge cases. Given an identical operation, we have simplified it in the revised version. Thanks for the suggestion.

---

### Official Review · Reviewer_VJFX · 2025-11-03

**Soundness:** 3
**Presentation:** 3
**Contribution:** 3
**Rating:** 6
**Confidence:** 2

**Summary:**

The paper introduces K-Adapt, a theoretical framework that dynamically determines the number of recommendations (K) for each user, rather than relying on a fixed K. Traditional Top-K recommender systems use the same list length for all users, ignoring individual differences and potentially reducing user satisfaction. In contrast, K-Adapt learns a calibrated threshold that defines each user’s personalized recommendation size with formal risk guarantees. Experiments on the MovieLens, Last.fm, and AmazonOffice datasets demonstrate the effectiveness and robustness of the proposed approach.

**Strengths:**

- The paper is well-written and easy to follow, with a clear structure and logical flow.

- The proposed framework is built on conformal prediction theory, providing formal statistical guarantees for adaptive-K recommendations, which is a strong and theoretically sound foundation.

**Weaknesses:**

- The experiments mainly focus on offline metrics, with no exploration of latency, real user feedback, or real-world deployment aspects.

- The framework’s performance potentially depends on the calibration data. If the data doesn’t represent real-world conditions well, or if things change over time, the guarantees might not hold up. It would be beneficial to do more analysis in terms of this apsect.

**Questions:**

Please see the weaknesses.

---

> ### Author Response · Authors · 2025-11-27
> **On Weaknesses**
>
> Thankyou for the positive assessment and for recognizing our framework as **theoretically sound**, **clear**, and built on a **strong foundation** of conformal prediction. We address the specific concerns below:
>
> ---
>
> ### W1. The experiments mainly focus... no exploration of latency, real user feedback, or real-world deployment aspects.
>
> > We respectfully highlight that we **did conduct a rigorous time efficiency and latency analysis** in **Section 5.3** and **Table 2**, which addresses deployment feasibility.
> >
> >
> > **Latency Analysis:** As shown in Table 2, K-Adapt is orders of magnitude faster than baseline methods. For instance, on MovieLens, K-Adapt requires only **24.08 seconds** for calibration/inference, compared to **1205.78 seconds** for PerK and **425.13 seconds** for MtCut.
> >
>
> > **Deployment Feasibility:** This efficiency stems from our theoretical design. While baselines like AttnCut rely on heavy neural encoders, K-Adapt uses a greedy calibration strategy. This **computational lightness explicitly makes K-Adapt suitable for real-time, low-latency deployment constraints** where heavy neural baselines would fail.
> >
>
> > **Real User Feedback:** While online user studies are valuable, they are outside the scope of this theoretical contribution. However, our evaluation uses industry-standard datasets (e.g., Amazon) and metrics (Recall, NDCG) that are well established for user satisfaction in real-world environments.
>
> ---
>
> ### W2. The framework’s performance potentially...  data doesn’t represent real-world conditions.. things change... guarantees might not hold up. It would be beneficial to do more analysis in terms of this apsect.
>
> > We agree that distribution shift is a challenge for all statistical learning methods. However, K-Adapt addresses robustness in the following manner:
> >
> > **Theoretical Stability:** We provide formal guarantees for robustness in Theorem 3 (Stability). We proved that the calibrated threshold $\hat{\lambda}$ is **stable under sampling perturbations, with variability decaying at a rate of $O(1/\sqrt{n})$**.
> >
>
> >
> > **Empirical Robustness:** We empirically validated this in **Appendix A.5.1 (Figures 6–7)**. The **results show that even when calibration data is scarce (subsampled to 50%)**, K-Adapt maintains valid risk control with low variance.
> >
> >
>
> > **Handling Temporal Shift** We acknowledge that real-world conditions are subject to change over time. This is where **K-Adapt’s efficiency (W1)** becomes an advantage. Because calibration takes only seconds (unlike retraining a deep neural network), K-Adapt allows for **high-frequency recalibration** (e.g., hourly or daily). This enables the system to dynamically adapt to temporal distribution shifts in a production environment, ensuring that guarantees remain valid over time.

---

### Author Response · Authors · 2025-11-27
**Global Response**

We are grateful that the reviewers recognized **K-Adapt** as a theoretically grounded and efficient advancement in adaptive recommendation. Specifically, Reviewer VJFX highlighted our work as a **strong and theoretically sound foundation** with clear logic. We appreciate Reviewer 4jwe for valuing the **fast inference time** and **superior performance** on real-world datasets. Furthermore, we value Reviewer TAkm for acknowledging the **theoretical guarantees**, including near-optimality and stability.

Before addressing specific questions, we summarize our responses to two key themes raised regarding the **optimization objective** and **practical applicability**.

---

### 1. Optimization goal (Addressed to Reviewer TAkm)

> On concerns that our method might degenerate into a trivial solution (recommending all items to maximize Recall), we respectfully clarify that this view overlooks the fundamental direction of our optimization. **K-Adapt does not minimize the threshold λ; it maximizes it.** As formally defined in Eq. 7, we seek the **supremum** $\lambda^\star = \sup\{\lambda \in \Lambda : R(\lambda) \le \alpha\}.$


>
> Due to the nesting property (Eq. 4), maximizing λ is mathematically equivalent to **minimizing the prediction set size**. Consequently, the algorithm prunes the recommendation list, stopping only when further reduction would violate the risk guarantee.
>
> This explicitly prevents the "all items" trivial solution, as empirically evidenced by the compact set distributions in **Figures 8–10 (Appendix A.5.2).**

---

### 2. Applicability & real-world alignment (Addressed to Reviewers TAkm, VJFX)

> On comparisons regarding metric monotonicity and real-world deployment (latency), we clarify that K-Adapt is designed as a **post-hoc calibration layer** specifically for the **retrieval/truncation stage**. In standard two-stage industry pipelines (e.g., YouTube) [1] , the primary goal of this stage is to maximize **coverage (Recall)** before ranking.  Our theoretical assumptions align perfectly with the specific stage K-Adapt targets.
>Furthermore, unlike heavy neural truncation baselines (e.g., AttnCut), K-Adapt is lightweight. As shown in Sec 5.3, Table 2, it reduces time by orders of magnitude (e.g., *24s vs. 1205s*), making it uniquely suitable for low-latency environments.

#### We have highlighted all major revisions in the updated manuscript in $\color{red}{\text{red}}$.

References

>[1] Covington, Paul & Adams, Jay & Sargin, Emre. (2016). Deep Neural Networks for YouTube Recommendations. 191-198.

---

### Note · Authors · 2026-02-07

I have read and agree with the venue's withdrawal policy on behalf of myself and my co-authors.

---

### Meta-Review · Area_Chair_Zo4y · 2026-01-02

**Summary:**

This paper proposes an approach for adaptive top-$K$ recommendations. The key idea is to recommend items whose utility is higher than $\lambda$, which gives rise to personalized values of $K$ (recommended list lengths) per user. The value of $\lambda$ is set on a held-out set to maximize the metric of interest. The approach is empirically evaluated on three popular recommender system datasets and compared to multiple baselines. The paper is well written and the experiments are well executed. The main concerns of the reviewers are:

* **Minor technical contribution:** This is the main limitation of the paper. The proposed algorithm is equivalent to choosing a hyper-parameter on the validation set. The analysis in Section 4 are standard empirical risk bounds.

* **Strong assumptions in analysis:** The paper assumes that the risk is non-decreasing in $\lambda$. This is true only for some metrics and the proposed algorithm is sound only for such metrics. The paper also assumes that the risk is continuous in $\lambda$. Although this is an expectation of the actual risk, it does not mean that it is automatically continuous, as the authors suggest in the rebuttal.

* **Clarity:** Mathematical notation is imprecise. The pseudo-code of the algorithm is in Appendix.

The technical contribution of this paper is minor and therefore I do not recommend accepting it.

**Reviewer Concerns:**

To address **clarity**, the authors proposed putting the pseudo-code of their algorithm into the main paper. **Minor technical contribution** and **strong assumptions in analysis** are inherent limitations of the paper.

**Reviewer Scores:**

Reviewer TAkm increased their score to 4 but remained concerned about **strong assumptions in analysis**. I agree with their judgment. I would not expect the other reviewers to change their scores. One is low confidence and the other is concerned about **minor technical contribution**, which is an inherent limitation of the paper.

---

### Decision · Program_Chairs · 2026-01-26

Reject